# Spatial predictive risk mapping of lymphatic filariasis residual hotspots in American Samoa using demographic and environmental factors

**Angela M. Cadavid Restrepo**[1]*, **Beatris M. Martin**[1], **Saipale Fuimaono**[2], **Archie C. A. Clements**[3], **Patricia M. Graves**[4], **Colleen L. Lau**[1]

1 School of Public Health, Faculty of Medicine, The University of Queensland, Brisbane, Queensland, Australia, 2 American Samoa Department of Health, Pago Pago, American Samoa, 3 Curtin School of Population Health, Faculty of Health Sciences, Curtin University, Perth, Western Australia, Australia, 4 College of Public Health, Medical and Veterinary Sciences, James Cook University, Cairns, Queensland, Australia

* a.cadavidrestrepo@uq.edu.au

**Data Availability Statement:** All relevant data are within the paper. We are unable to provide individual-level antigen and antibody prevalence

## Abstract

### Background

American Samoa successfully completed seven rounds of mass drug administration (MDA) for lymphatic filariasis (LF) from 2000–2006. The territory passed the school-based transmission assessment surveys in 2011 and 2015 but failed in 2016. One of the key challenges after the implementation of MDA is the identification of any residual hotspots of transmission.

### Method

Based on data collected in a 2016 community survey in persons aged ≥8 years, Bayesian geostatistical models were developed for LF antigen (Ag), and Wb123, Bm14, Bm33 antibodies (Abs) to predict spatial variation in infection markers using demographic and environmental factors (including land cover, elevation, rainfall, distance to the coastline and distance to streams).

### Results

In the Ag model, females had a 26.8% (95% CrI: 11.0–39.8%) lower risk of being Ag-positive than males. There was a 2.4% (95% CrI: 1.8–3.0%) increase in the odds of Ag positivity for every year of age. Also, the odds of Ag-positivity increased by 0.4% (95% CrI: 0.1–0.7%) for each 1% increase in tree cover. The models for Wb123, Bm14 and Bm33 Abs showed similar significant associations as the Ag model for sex, age and tree coverage. After accounting for the effect of covariates, the radii of the clusters were larger for Bm14 and Bm33 Abs compared to Ag and Wb123 Ab. The predictive maps showed that Ab-positivity was more widespread across the territory, while Ag-positivity was more confined to villages in the northwest of the main island.

data and demographic data because of the potential for breaching participant confidentiality. The communities in American Samoa are very small, and individual-level data such as age, sex, and village of residence could potentially be used to identify specific persons. For researchers who meet the criteria for access to confidential data, the data are available on request from the Human Ethics Officer at the Australian National University Human Research Ethics Committee, email: human. ethics.officer@anu.edu.au. Protocol number 2016/ 482.

**Funding:** This study was supported by the Coalition for Operational Research on Neglected Tropical Diseases (COR-NTD), which is funded at The Task Force for Global Health primarily by the Bill & Melinda Gates Foundation [OPP1053230 to CLL], the United Kingdom Department for International Development, and by the United States Agency for International Development through its Neglected Tropical Diseases Program. CLL was supported by Australian National Health and Medical Research Council Fellowships (APP1193826). The funders had no role in study design, data collection and analysis, decision to publish, or preparation of the manuscript.

**Competing interests:** The authors have declared that no competing interests exist.

## Conclusion

The findings may facilitate more specific targeting of post-MDA surveillance activities by prioritising those areas at higher risk of ongoing transmission.

## Author summary

The Global Programme to Eliminate Lymphatic filariasis (LF) aims to interrupt transmission by implementing mass drug administration (MDA) of antifilarial drugs in endemic areas; and to alleviate suffering of those affected through improved morbidity management and disability prevention. Significant progress has been made in the global efforts to eliminate LF. One of the main challenges faced by most LF-endemic countries that have implemented MDA is to effectively undertake post-validation surveillance to identify residual hotspots of ongoing transmission. American Samoa conducted seven rounds of MDA for LF between 2000 and 2006. Subsequently, the territory passed transmission assessment surveys in February 2011 (TAS-1) and April 2015 (TAS-2). However, the territory failed TAS-3 in September 2016, indicating resurgence. We implemented a Bayesian geostatistical analysis to predict LF prevalence estimates for American Samoa and examined the geographical distribution of the infection using sociodemographic and environmental factors. Our observations indicate that there are still areas with high prevalence of LF in the territory, particularly in the north-west of the main island of Tutuila. Bayesian geostatistical approaches have a promising role in guiding programmatic decision making by facilitating more specific targeting of post-MDA surveillance activities and prioritising those areas at higher risk of ongoing transmission.

## Introduction

Lymphatic filariasis (LF) is a vector-borne parasitic disease caused by three species of filarial worms–*Wuchereria bancrofti*, *Brugia malayi*, and *B. timori* [1]. The presence of adult worms in the lymphatic vessels leads to damage of the lymphatic system, causing clinical disease characterised by lymphoedema of the limbs or genitals, such as elephantiasis and scrotal hydrocoeles [1]. LF is one of the leading causes of chronic disability worldwide, being responsible for over 5 million disability-adjusted life years before the implementation of elimination strategies against the infection [2,3].

In 1997, the World Health Organization (WHO) targeted LF for global elimination as a public health problem by 2020 [4]. Subsequently, WHO launched the Global Programme to Eliminate Lymphatic Filariasis (GPELF) in 2000 that included two strategies: first, the implementation of mass drug administration (MDA) to interrupt the community-level transmission of LF, and second, management and prevention of morbidity and disability for people with chronic complications [5]. By 2019, 72 countries were still considered endemic by the GPELF and 50 still required MDA [6]. A number of countries have already achieved validation of LF elimination as a public health problem after intensive community-based MDA programs (including Cambodia, The Cook Islands, Egypt, Kiribati, Malawi, Maldives, Marshall Islands, Niue, Palau, Sri Lanka, Thailand, Togo, Tonga, Vanuatu, Viet Nam, Wallis and Futuna, and Yemen) [6]. Some countries have stopped MDA and are under surveillance to determine if LF elimination criteria have been met. One of the main challenges faced by most LF-endemic

countries that have implemented MDA is to effectively undertake post-MDA and post-validation surveillance [7].

American Samoa successfully completed seven rounds of MDA with a single dose of diethylcarbamazine (DEC) and albendazole from 2000–2006. Subsequently, the territory passed the WHO-recommended school-based transmission assessment surveys (TAS) conducted in 2011 (TAS-1) and 2015 (TAS-2) with crude prevalences of antigen (Ag)-positive of 0.2% (95% confidence interval (CI) 0.0 to 0.8%) and 0.1% (95% CI 0.0 to 0.7%), respectively [8,9]. Despite this achievement, the territory failed TAS-3 in 2016 with an adjusted Ag prevalence of 0.7% (95% CI 0.3 to 1.8%), higher than the threshold and the recommended upper confidence limit of 1% [10]. The findings in TAS-3 suggested potential resurgence of LF in the territory and were confirmed by a community-based survey conducted in the same year with an Ag prevalence of 6.2% (95%CI 4.5 to 8.6%) in individuals aged ≥8years [10]. Evidence from others studies conducted in the territory in 2010 and 2014 also suggested ongoing LF transmission and the potential persistence of residual hotspots [8,9,11,12].

WHO recommends conducting follow-up surveys of nearby households of Ag-positive children identified through TAS to complement post-MDA surveillance [13]. However, the recommendations are vague and lack a clear threshold for triggering a programmatic response [14]. As LF prevalence decreases, the ability of diagnostic methods that are sufficiently sensitive, particularly in the TAS, to detect areas with residual transmission is also limited [7]. This limitation is of particular importance in areas where the geographical distribution of LF has been demonstrated to be highly heterogeneous [15]. Currently, TAS relies on the monitoring of antigenemia in children aged 6 to 7 years [16]; however, since Ag specific antifilarial antibodies such as Wb123, Bm14 and Bm13, generally develop before patent infection and may be indicators of different infection stages (LF exposure or infection patterns) their use in post-MDA surveillance could help to provide an early measure of filarial exposure and ongoing transmission [17].

In American Samoa, a recent study confirmed clustering of the infection in areas that were previously suspected as hotspots in 2010 and 2014 (Fagali'i village in the far north-west of Tutuila island, and also in the Ili'Ili-Vaitogi-Futiga area that is located on the south coast) and identified other potential areas where there is still potential residual infection [8,18]. Therefore, strategies for identifying foci of infection in low-prevalence settings are crucial in the context of the LF elimination efforts, both from the perspective of targeting communities for MDA and also for understanding the future of the post-MDA surveillance needs.

*W. bancrofti*, *B. malayi*, and *B. timori* require two hosts to complete its life cycle, the human and the mosquito hosts. Therefore, sociodemographic, economic and environmental factors that act at different spatial scales have the potential to influence the transmission pathways of the parasites [19]. The clustered distribution of LF has been associated with landscape characteristics and climatic factors in several LF-endemic areas including the Pacific Islands [20,21]. Bayesian model-based geostatistics incorporating infection prevalence data with socio-demographic and environmental covariates has proven to be able to predict disease distribution in areas with scarce information [22–24]. Hence, understanding how environmental and socio-demographic factors interact to determine parasite transmission is essential for the design and implementation of effective elimination strategies against LF.

The aim of this study was to identify areas where there is potential residual transmission of *W. bancrofti* in American Samoa and produce LF predictive prevalence maps that can be used to help guide and target future LF elimination strategies. A Bayesian model-based geostatistics approach was used to: (i) assess and quantify the associations between LF infection markers and sociodemographic and environmental factors at the household level and (ii) develop spatial prediction of prevalence estimates of LF in American Samoa using different infection markers–Ag and antibodies (Abs) against Wb123, Bm14, Bm33.

## Methods

### Ethics statement

Ethical approval for the 2016 field survey was obtained from the American Samoa Institutional Review Board and the Human Research Ethics Committee at the Australian National University (protocol number 2016/482) and the University of Queensland (2021/HE000896). After explaining the purpose and procedures of the survey, all adults and parents/guardians of the minors (<18 years) who agreed to participate were asked to sign an informed written consent form. Full details of local collaborations and official permissions to visit villages have been previously described [10].

### Study area

American Samoa is a United States territory in the South-central Pacific located approximately between latitudes 11° North and 15° South and longitudes 168° East and 172° West (Fig 1). The total land area of the territory is 200 km² and comprises five inhabited volcanic islands Tutuila, Aunu'u, Ofu, Olosega and Ta'ū, and two remote coral atolls (Swains Island and Rose Atoll). In 2010, the population of American Samoa was 55,519, the majority of whom (95%) lived in Tutuila, the largest island (198.9 km²), where the capital Pago Pago is located [25].

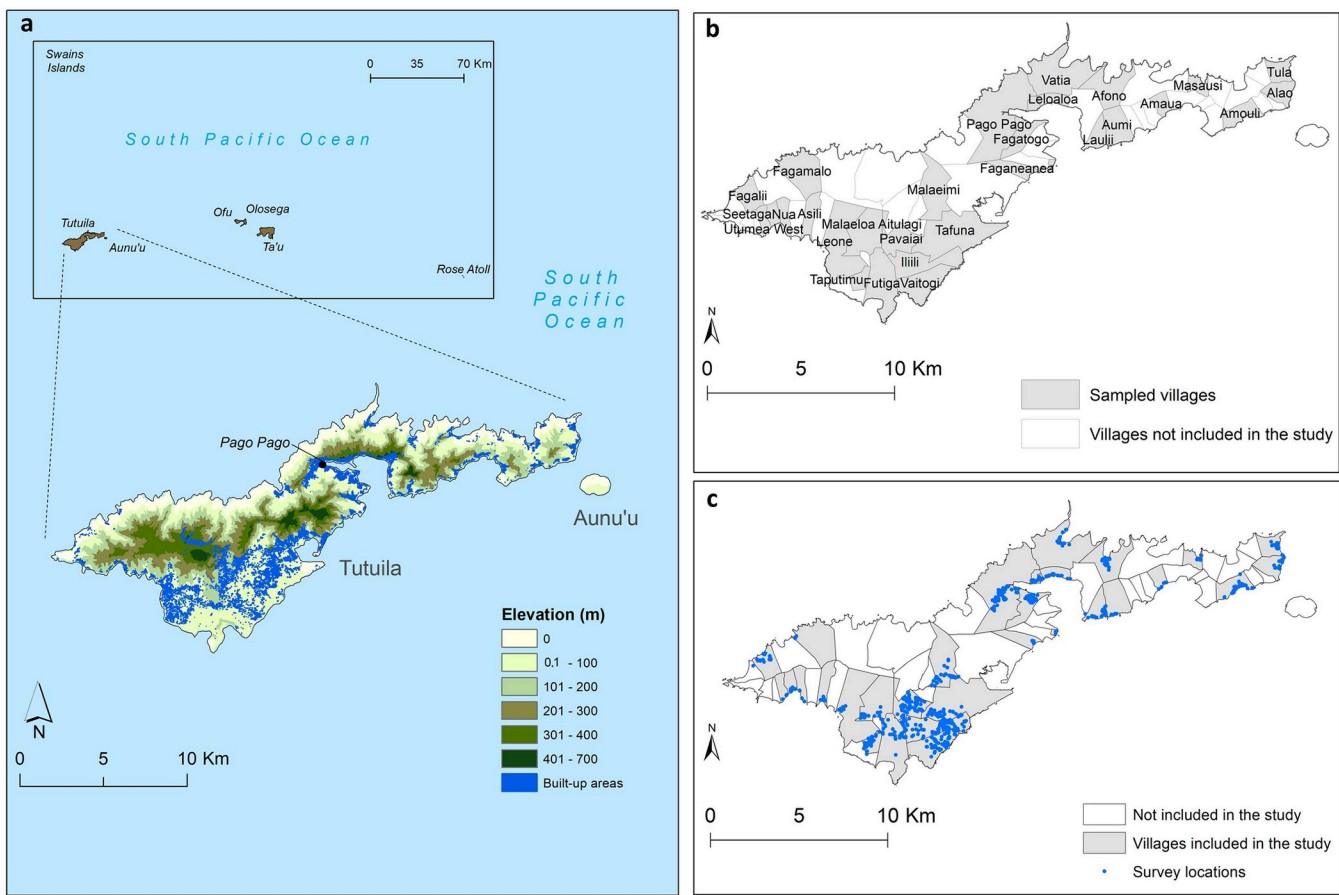

**Fig 1.** Map of American Samoa (a) Elevation and distribution of the built-up areas in the main island of Tutuila, (b) distribution of selected villages and (d) survey locations. Base layers from: (https://www.diva-gis.org/gdata), (http://pacioos.org/metadata/as_dw_tau_bldngs.html) and (http://pacioos.org/metadata/usgs_dem_10m_tutuila.html).

American Samoa lies in the tropical savanna climate zone characterized by alternate wet (October to May) and dry (June to September) seasons [26]. Temperatures vary slightly between the hottest period (December to April), when the average is approximately 31˚C, and the coolest period (June to August), when the average is 29˚C. The annual average rainfall ranges from 3000 to 6000 mm, with 70% occurring during the hot and wet season [26]. The average elevation is 482 meters (m) with the highest point being Lata Mountain on the island of Ta'ū (970 m) [27].

## Data from community survey of lymphatic filariasis in 2016

Data on LF infection markers, Ag and Wb123, Bm14 and Bm33 Abs, were obtained from a two-stage equal probability cluster survey conducted in American Samoa in 2016. Full details about survey design and sampling methods have been previously reported [10]. Briefly, 30 primary sampling units (PSUs) were randomly selected from a total of 70 villages/village segments/village groups, that were defined based on a population size of less than 2000. Two villages that were previously identified and confirmed as LF hotspots in 2010 and 2014, respectively, were also added to the survey as PSUs [8]. Based on a population of 55,000 [25], a target sample size of 4,620 was estimated to be required to detect Ag prevalence of 1% for persons aged ≥8 years. The target number of households that needed to be visited was calculated by dividing the target sample size of persons by the average household size and accounting for a 15% non-response/ absentee rate (the most recent census provided the total numbers of households in the selected villages and indicated that the average of persons per household was seven). A sampling fraction of 0.29 was calculated as the proportion of households that needed to be sampled on each PSU to achieve the target sample size. However, after the first two weeks of recruitment, the observed antigen prevalence was approximately 4%, which is significantly higher than anticipated, and it was determined that a smaller target sample size of 2,981 would be sufficient to obtain adequate statistical power. Within each PSU, a population proportionate sampling method was implemented to randomly select households from a geo-referenced list of buildings obtained from the American Samoa Department of Commerce [27]. In total, the survey included 32 PSUs (across 30 villages) and 754 households.

A household member was defined as an individual who considered the selected house as their principal place of residence or who slept in that house the previous night. All consenting household members aged ≥8 years were surveyed and blood samples were tested for circulating filarial Ag using the Alere Filariasis Test Strip (FTS) (Abbott, Scarborough, ME) [28] and for Wb123, Bm14 and Bm33 Abs using multiplex bead assays (MBA) [29].

Standardised electronic questionnaires were administered by bilingual field research assistants (in Samoan or English based on each participant's preference). The demographics data collected included sex, age and work location. Work location was categorised as indoor, outdoor, tuna cannery (largest private employer in American Samoa), and other (including mixed indoor/outdoor, unemployed, retired or unknown).

## Geospatial data sources

We downloaded and assembled spatial and environmental data that have been found to be associated with the geographical distribution of LF in other endemic regions [19,23,30–32]. The boundary administrative maps and the covariate data consider for the analyses were derived from the following datasets:

i. *Village boundaries and building footprints.* Maps of village boundaries and building footprints were downloaded from DIVA-GIS (https://www.diva-gis.org/gdata/) and the Pacific

Islands Ocean Observing System (PacIOOS) websites (https://www.pacioos.hawaii.edu/metadata/as_dw_tut_bldngs.html) [33,34].

ii. *Coastline and streams*. The American Samoa coastline and network of streams covering the entire territory were extracted in a shapefile format from the Pacific Islands Ocean Observing System (PacIOOS) website (https://www.pacioos.hawaii.edu/data/search-results/?text=streams%20american%20samoa) [35].

iii. *Population density*. Data on population density for 2010/2011 were downloaded from the Pacific Data Hub website (https://pacificdata.org/data/dataset/asm_population_grid_2020) [36]. A grid (i.e. raster surface) was available for American Samoa at the resolution of 100 m.

iv. *Elevation*. Data were obtained in a GeoTIFF format at the spatial resolution of 10 m from the United States Geological Survey (USGS) 10-m Digital Elevation Model (DEM): American Samoa: Tutuila (http://pacioos.org/metadata/usgs_dem_10m_tutuila.html) [37].

v. *Rainfall*. Average monthly rainfall for 2016 were downloaded from the Pacific Environment Data Portal (https://pacific-data.sprep.org/) [38] in a raster format at the spatial resolution of 1 km. There was limited availability of spatial monthly rainfall datasets for the years prior to the survey. Therefore, the monthly rainfall layers from 2016 were used based on the assessment of the representativeness of the ten-year period prior the survey (S1 Table and S1 and S2 Figs).

vi. *Land surface temperature*. Satellite sensor data on land surface temperatures from the Moderate Resolution Imaging Spectroradiometer (MODIS) satellite were obtained from the USGS Earth Explorer website (https://lpdaac.usgs.gov/products/mod11a2v006/) [39]. These data were downloaded at 1 km resolution for every eight days from January 1 to December 31 2016.

vii. *Land use/land cover map*. Data were derived at 10m resolution from the Sentinel-2 Global Land Use/Land Cover (LULC) Timeseries produced by Impact Observatory, Microsoft, and the Environmental Systems Research Institute (Esri) (https://www.arcgis.com/apps/instant/media/index.html?appid=fc92d38533d440078f17678ebc20e8e2) [40].

## Covariate data download and processing

The geo-referenced data sets that included the locations of the surveyed households, the covariates and the boundary map of American Samoa were imported into ArcGIS version 10.7.1 [41] to extract data (measured on a continuous scale) for the territory. The geographical distributions of the covariates are shown in Fig 2.

- Elevation estimates for the territory were extracted in meters (m) above sea level.

- A layer of the distance between each household location and the nearest coastline was developed (in m) using the Euclidean Distance Tool.

- The Euclidean Distance Tool was also used to estimate the distance (in m) between each household location and the nearest permanent surface stream.

- The monthly rainfall (mm) datasets were used to estimate the annual average rainfall and rainfall of the driest (August) and wettest (December) months in 2016.

- Annual average temperature and temperature of the hottest (December) and coolest (July) months in 2016 were estimated from the fortnightly temperature layers.

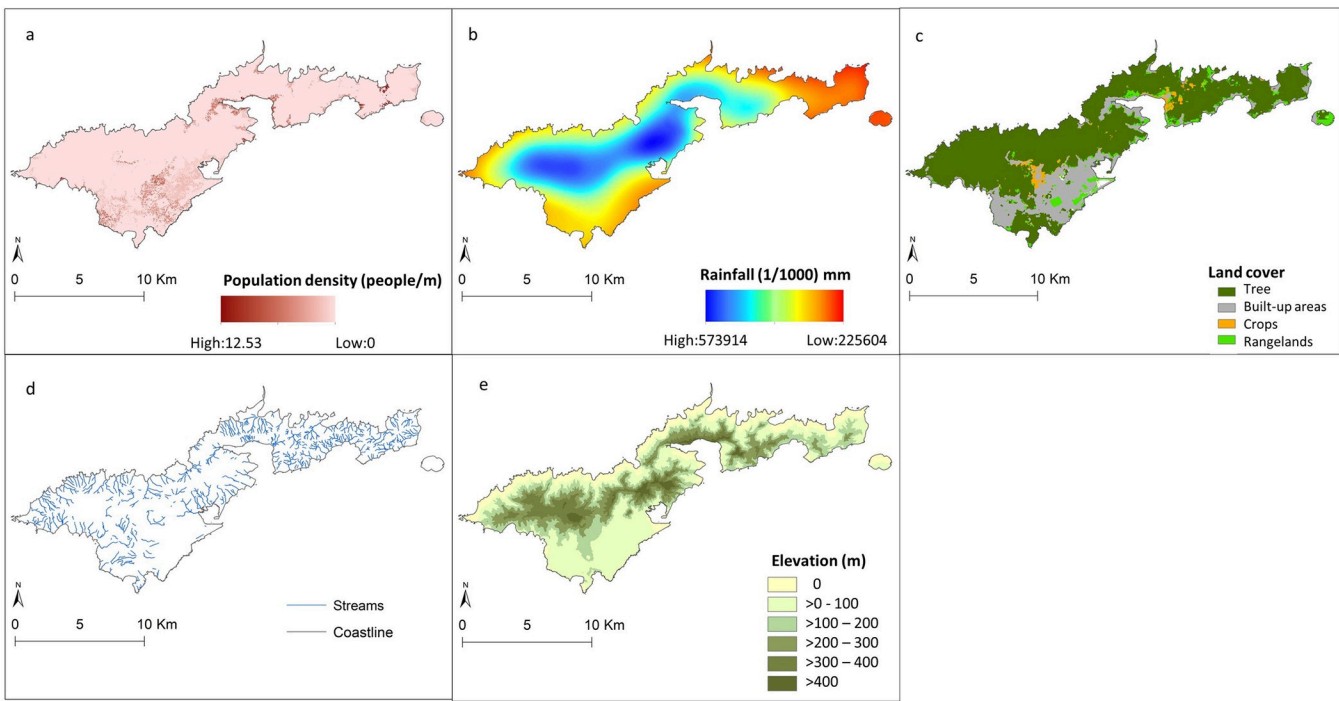

**Fig 2.** The geographical distributions of the covariates in American Samoa (a) Population density (people/m2) (https://pacificdata.org/data/dataset/asm_population_grid_2020), (b) Annual average rainfall (mm) (https://pacific-data.sprep.org/), (c) Land cover (https://www.arcgis.com/apps/instant/media/index.html?appid=fc92d38533d440078f17678ebc20e8e2), (d) streams and coastline (https://www.pacioos.hawaii.edu/data/search-results/?text=streams%20american%20samoa) and (e) Elevation (m) (http://pacioos.org/metadata/usgs_dem_10m_tutuila.html). Base layer from: (https://www.diva-gis.org/gdata/).

- The global LULC cover map with 11 LULC classes was used to generate four separate rasters for the LULC categories that cover the territory of American Samoa: crops, rangelands, trees, and built/urban area (Table 1).

**Table 1. Land cover class definitions.**

| Land Use/Land cover Class | Description |
|---|---|
| Crops | Human planted/plotted cereals, grasses, and crops not at tree height. Examples: corn, wheat, soy, fallow plots of structured land. |
| Rangeland | Open areas covered in homogenous grasses with little to no taller vegetation; wild cereals and grasses with no obvious human plotting (i.e., not a plotted field). Examples: natural meadows and fields with sparse to no tree cover, open savanna with few to no trees, parks/ golf courses/lawns, pastures. Mix of small clusters of plants or single plants dispersed on a landscape that shows exposed soil or rock; scrub-filled clearings within dense forests that are clearly not taller than trees; examples: moderate to sparse cover of bushes, shrubs and tufts of grass, savannas with very sparse grasses, trees or other plants. |
| Trees | Any significant clustering of tall (~15 feet or higher) dense vegetation, typically with a closed or dense canopy. Examples: wooded vegetation, clusters of dense tall vegetation within savannas, plantations, swamp or mangroves (dense/tall vegetation with ephemeral water or canopy too thick to detect water underneath). |
| Built/Urban | Human made structures; major road and rail networks; large homogenous impervious surfaces including parking structures, office buildings and residential housing. Examples: houses, dense villages / towns / cities, paved roads, asphalt. |

## Buffer zones

The GPS locations of the surveyed households were used to delineate a buffer zone of 20 m around the households in ArcGIS [41]. The buffer size was selected to represent an approximate distance within which the participants would spend extensive periods of time, and therefore have greatest exposure to the environmental conditions within the buffers [42]. For each surveyed location, the data extracted within the buffer zone included the spatial mean values of population density, distance to coastline and streams, elevation, annual average rainfall, rainfall in the wettest (December) and driest (August) months in 2016, annual average temperature, and temperature of hottest (January) and coolest (July) months in the same year. Each of the four land cover classes covering the American Samoa territory were summarised as percentages of area within the 20 m buffer.

## Descriptive analyses

For each infection marker and the covariates, summary statistics were calculated in R software R-4.0.3 [43]. Crude prevalence of Ag, and Wb123, Bm14 and Bm33 Abs were estimated and mapped at the village level, and binomial exact methods were applied to estimate 95% confidence intervals (95% CI). Of note, in all subsequent analyses data were examined at the individual level and the respective household locations.

## Variable selection

Collinearity between covariates was assessed using Spearman's correlation. Non-spatial univariate logistic regression models were developed using R software R-4.0.3 [43] to examine the association of each LF infection marker (outcome variables) with the sociodemographic and environmental factors (covariates). For the strongly correlated covariates (Spearman's correlation coefficient $\rho > 0.8$), the ones with the highest value of Akaike Information Criterion (AIC) in the univariate regression models were excluded (S3 Fig). For each infection marker, multivariate logistic regression models were developed incorporating the remaining covariates. From these models, covariates were sequentially removed to assess AIC and p-values. Nonlinear associations between predictors and the outcome variables were modelled using quadratic terms. The models with the lowest AIC were selected for further analyses and covariables with p <0.05 were retained.

## Multivariable non-spatial and spatial regression models

Bayesian geostatistical multivariate regression models were fitted using the OpenBUGS software version 3.2.3 rev 1012 [44]. For each infection marker, separate logistic regression models were developed based on the binary outcome of the laboratory results. First, non-spatial models were developed with the sociodemographic and environmental covariates as fixed effects but without considering the spatial dependence of the data. Then, geostatistical models for each infection marker were fitted using a Markov chain Monte Carlo (MCMC) simulation approach with Gibbs sampling (S1 Text) [45].

The deviance information criterion (DIC) statistic was calculated to assess if the inclusion of spatial dependence in the data improved the fit of the models. Low DIC values indicate a better fit. Covariates in the models were considered statistically significant if the 95% credible intervals (95% CrI) of the estimated odds ratios (OR) excluded 1.

The mathematical notation of the spatial model is provided below, and contains all of the components of the non-spatial model. Assuming a Bernoulli-distributed dependent variable, $Y_{ij}$, corresponding to the results of the infection markers (0 = negative, 1 = positive) of the $i$th

participant ($I = 1 \ldots 2{,}671$) the $j$th location ($j = 1 \ldots 736$), the model structures were as follows:

$$Y_{ij} \sim Bern(p_{ij})$$

$$\mathrm{logit}(p_{ij}) = \alpha + \gamma \times \mathrm{age}_i + \delta \times \mathrm{female}_i + \varepsilon \times \mathrm{outdoor}_i + \eta \times \mathrm{tuna\ cannery}_i + \theta \times \mathrm{others}_i$$

$$+ \sum_{z=1}^{z} \beta_z \times \lambda_{zj} + s_j$$

where $\alpha$ is the intercept, $\gamma$ and $\delta$ are coefficients for age and females, and $\varepsilon$, $\eta$ and $\theta$ are coefficient for the occupation categories. $\beta$ is a matrix of z coefficients, $\lambda$ is a matrix of z environmental variables and population density, and $s_j$ a geostatistical random effect. The correlation structure of the geostatistical random effect was assumed to be an exponential function of the distance between points:

$$f(d_{kl}; \phi) = \exp[-\phi d_{kl}]$$

where $d_{kl}$ are the distances between pairs of points $k$ and $l$, and $\phi$ is the rate of decline of spatial correlation per unit of distance. A normal distribution was used for the priors for the intercept and the coefficients (mean = 0 and precision, the inverse of variance, $= 1 \times 10^{-3}$), whereas a uniform distribution was specified for $\phi$ (with upper and lower bounds s = 0.03 and 100; the lower bound set to ensure spatial correlation at the maximum separating distance between survey locations was <0.5). A non-informative gamma distribution was used to specify the priors for the precision (shape and scale parameters = 0.001, 0.001).

A burn-in of 1,000 iterations were run first and discarded. Sets of 10,000 iterations were then run and examined for convergence. Convergence was assessed by visual inspection of history and density plots and by examining autocorrelation of the model parameters. In each model, convergence was achieved for all variables at approximately 30,000 iterations. The last 10,000 values from the posterior distributions of each model parameter were recorded (S4, S5, S6, and S7 Figs). The rate of decay of spatial correlation between locations ($\phi$) with distance and the variance of the spatial structured random effect ($\sigma^2$) were also stored.

## Predicted prevalence of lymphatic filariasis and model validation

To predict LF prevalence at unsampled locations, a regular 150 m × 150 m grid was overlaid on a map of American Samoa to extract the average environmental data for each grid cell. The predicted probabilities at the unsampled locations were estimated using the *spatial.unipred* function in OpenBUGS. The function applies the model equation at each unsampled location using the covariates values extracted for them and the distance between those locations and the surveyed locations. Bayesian kriging was applied in ArcGIS to generate smooth risk maps of the posterior distributions of predicted prevalence of each LF infection marker.

To determine the predictive performance of the models of predicted probability of each infection marker, a validation dataset was created by random selection of 25% of the data. The ability of the final models to predict the probability of Ag, Wb123, Bm14 and Bm33 Abs was assessed by comparing the predicted probability of each infection marker to the observed results (0 = negative, 1 = positive) in the validation locations. The area under curve (AUC) statistic of the receiver operating characteristic curve was used to quantify the discriminatory performance of the models [46]. AUC values <0.50 indicate that the model does not predict any more successfully than random allocation of test status; values of 0.50–0.69 indicate poor predictive performance; values of 0.70–0.89 have reasonable predictive performance and values >0.90 indicate a very good predictive performance. [46].

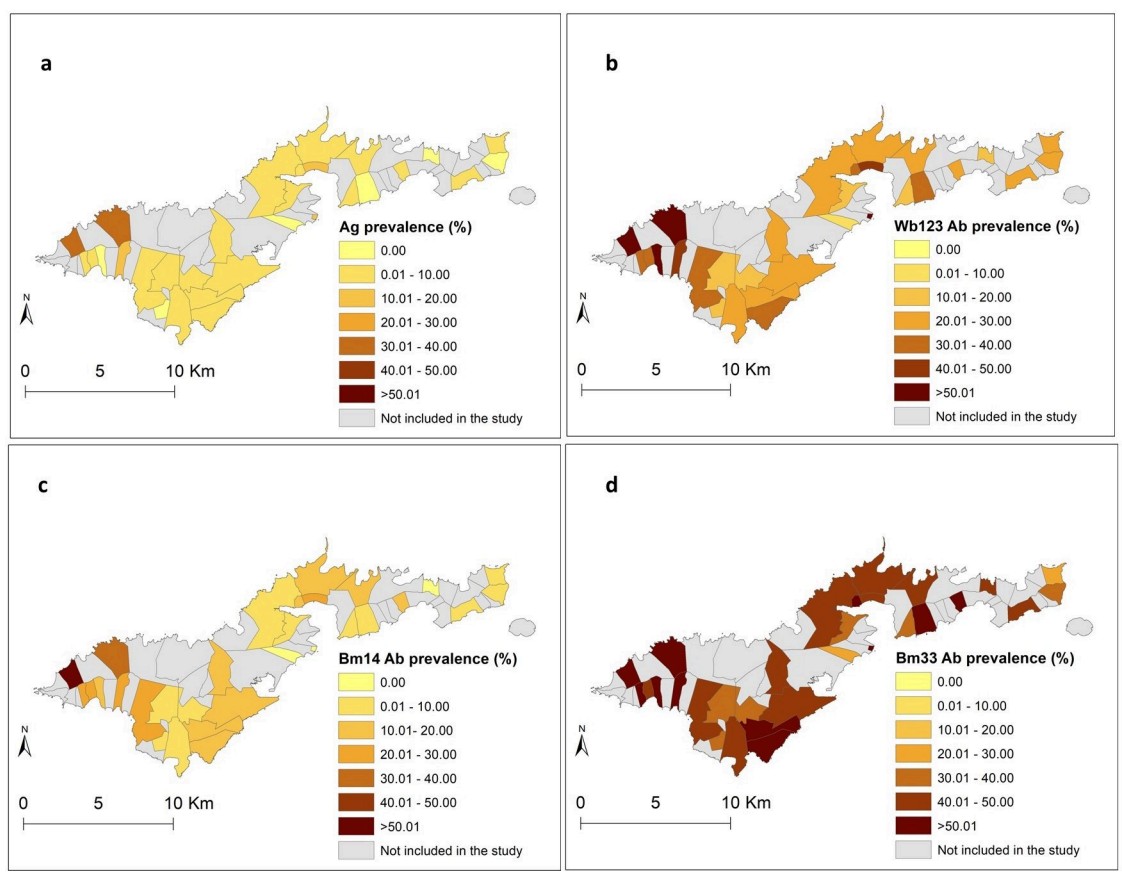

**Fig 3.** a) Geographical distribution of the prevalence of a) Ag, b) Wb123 Ab, c) Bm14 Ab and d) Bm33 Ab. Base layer from: (https://www.diva-gis.org/gdata/).

## Results

### Sample description and sample site locations

The final dataset used for analyses included 754 households in 736 unique locations (some households shared the same building structure) from 32 PSUs in 30 villages. The total number of participants was 2,671 with a mean age of 33.5 years (range 8–93), and 54.7% (n = 1462) were female. Fig 1B and 1C show the locations of sampled villages and the geographical distribution of the survey locations, respectively. The highest overall crude prevalence was observed for Bm33 Ab (45.6%, 95% CI 43.7−47.5%), followed by Wb123 Ab (25.6%, 95% CI 24.0 − 27.3%), Bm14 Ab (13.1%, 95% CI 11.8−14.4%) and Ag (5.1, 95% CI 4.2−5.9%). At the village level, Fagali'i (n = 81) and Fagamalo (n = 13), located in the far north-west of Tutuila Island, consistently showed high overall crude prevalence of all infection markers. A detailed description of the Ag and Ab results has been presented elsewhere [7,10,18]. Fig 3A, 3B, 3C and 3D display the observed geographical distributions of the prevalence of Ag and Wb123, Bm14 and Bm33 Abs, respectively, by village. The maps confirm that villages with high prevalence of Bm33 Ab were more widespread across the territory, while the distribution of villages with high prevalence of Ag, Wb123 and Bm14 Ab was more confined to the north-west of the territory.

**Table 2. Descriptive statistics of environmental covariates within 20 m buffers of surveyed household locations in American Samoa in 2016.**

| Variable | Mean | Median | Standard deviation | Minimum | Maximum |
|---|---|---|---|---|---|
| Population density (people/m$^2$) | 2.47 | 2.26 | 2.06 | 0.01 | 12.53 |
| Elevation (m) | 77.72 | 43.96 | 92.99 | 0 | 479.86 |
| Distance to the coastline (m) | 1081.97 | 644.25 | 1117.78 | 3.06 | 4371.57 |
| Distance to streams (m) | 358.25 | 151.12 | 434.39 | 0.32 | 1825.00 |
| Average annual rainfall (mm) | 3469.69 | 3374.57 | 698.28 | 2095.10 | 4630.68 |
| Rainfall in the driest month—August (mm) | 232.46 | 232.97 | 51.99 | 125.27 | 324.61 |
| Rainfall in the wettest month—December (mm) | 381.14 | 369.73 | 60.28 | 255.96 | 528.90 |
| Land cover | | | | | |
| Cropland (%) | 0.01 | 0 | 0.03 | 0 | 0.91 |
| Rangeland (%) | 0.85 | 0 | 6.83 | 0 | 0.99 |
| Tree coverage (%) | 5.14 | 0 | 19.93 | 0 | 99.00 |
| Urban (%) | 76.36 | 100 | 34.38 | 0 | 100.00 |

### Variable selection and univariate regression analyses

The descriptive statistics of the covariates considered for the analyses are presented in Table 2. Because temperature data were not available for large areas of American Samoa, this covariate was excluded from analyses. We identified four pairs of variables with Spearman's rank >0.8 that were assessed with the univariate regression models. The quadratic terms did not improve model fit and were not included in the final models. After comparing the AIC of the stepwise multivariate logistic regression models, the selected variables for the Bayesian non-spatial and spatial analyses included: sex, age, work location, population density, elevation, rainfall in the wettest month (December), distance to streams, cropland, tree coverage and urban areas.

### Bayesian non-spatial and spatial models

For all infection markers, the best-fit model included the spatial random effect. Tables 3 and S2 show the odds ratios (ORs) and 95% CrI from the Bayesian non-spatial and spatial models for Ag and Wb123, Bm14 and Bm33 Abs, respectively.

### Multivariate non-spatial and spatial models for Ag

The DICs of the models of Ag with and without accounting for spatial correlation were 631.70 and 1122.3, respectively. In the spatial model, females had a 26.8% (95% CrI: 11.0–39.8%) lower risk of being Ag-positive than males. There was a 2.4% (95% CrI: 1.8–3.0%) decrease in the odds of Ag positivity for every year of age. Tree coverage was also positively associated with Ag-positivity, with an estimated increase of 0.4% (95% CrI: 0.1–0.7%) in the odds of Ag-positivity for each 1% increase in the extent of tree coverage in the 20 m buffers.

After accounting for the effect of the statistically significant variables, the variance of the spatially structured random effect was 1.9 (0.9 to 2.9). The values of the decay parameter for spatial correlation $(\phi)$, was 77.1. This means that, after accounting for the effect of covariates, the radius of the clusters was approximately 4.3 km. ($\phi$ is measured in decimal degrees, therefore, the cluster size is calculated dividing 3 by $\phi$; at the equator, one decimal degree is approximately 111 km).

### Multivariate geostatistical model for Wb123, Bm14 and Bm33 Abs

The DICs of the models of positivity for Wb123 Ab with and without accounting for spatial correlation were 1996 and 2861.1, respectively. In the spatial model, females had a 55.1% (95%

**Table 3. Odd ratios (ORs) and 95% credible interval (CrI) from the Bayesian geostatistical models for Ag and Wb123, Bm14 and Bm33 antibodies in the community survey in American Samoa in 2016.** Statistically significant ORs are highlighted in blue (positive associations) and grey (negative associations).

| Model | Participants N (%) | Antigen positive | Wb123 antibody positive | Bm14 antibody positive | Bm33 antibody positive |
|---|---|---|---|---|---|
| | | ORs, posterior mean (95% CrI) | ORs, posterior mean (95% CrI) | ORs, posterior mean (95% CrI) | ORs, posterior mean (95% CrI) |
| **Gender** | | | | | |
| Male | 1209 (45.26) | Ref | Ref | Ref | Ref |
| Female | 1462 (54.74) | 0.22 (0.13 to 0.36) | 0.45 (0.36 to 0.56) | 0.48 (0.35 to 0.64) | 0.73 (0.60 to 0.89) |
| **Age (per year)** | - | 1.04 (1.02 to 1.05) | 1.02 (1.02 to 1.03) | 1.03 (1.03 to 1.04) | 1.02 (1.02 to 1.03) |
| **Work location** | | | | | |
| Indoor | 727 (27.22) | Ref | Ref | Ref | Ref |
| Outdoor | 40 (1.50) | 1.94 (0.56 to 6.22) | 1.49 (0.66 to 3.44) | 2.56 (1.70 to 6.59) | 2.62 (1.09 to 6.35) |
| Tuna cannery | 131 (4.90) | 1.05 (0.87 to 1.27) | 2.26 (1.37 to 3.77) | 2.56 (1.40 to 4.62) | 1.56 (1.07 to 2.50 |
| Others | 1773 (66.40) | 1.34 (0.79 to 2.42) | 1.43 (1.07 to 1.93) | 1.20 (0.82 to 1.72) | 1.08 (0.83 to 1.44) |
| **Population density (people/m$^2$)** | - | 0.89 (0.72 to 1.11) | 0.96 (0.86 to 1.06) | 0.92 (0.80 to 0.99) | 0.96 (0.88 to 1.05) |
| **Elevation (m)** | - | 1.00 (0.99 to 1.00) | 1.00 (0.99 to 1.01) | 1.00 (0.99 to 1.00) | 1.00 (0.99 to 1.00) |
| **Distance to streams (m)** | - | 1.00 (0.99 to 1.00) | 1.00 (0.99 to 1.00) | 0.99 (0.99 to 1.00) | 0.99 (0.99 to 1.00) |
| **Rainfall in the wettest month—December (mm)** | - | 1.00 (0.99 to 1.01) | 1.00 (0.99 to 1.01) | 1.00 (0.99 to 1.01) | 1.00 (0.99 to 1.00) |
| **Land Cover** | | | | | |
| Cropland (%) | - | 0.00 (0.00 to 1.04) | 1.06 (0.97 to 1.15) | 0.96 (0.80 to 1.06) | 1.05 (0.99 to 1.15) |
| Trees (%) | - | 1.00 (0.99 to 1.00) | 1.00 (1.001 to 1.01) | 1.01 (1.001 to 1.01) | 1.01 (1.001 to 1.01) |
| Built/Urban (%) | - | 1.00 (0.99 to 1.00) | 1.00 (0.99 to 1.00) | 0.99 (0.99 to 1.00) | 0.99 (0.99 to 1.00) |
| Heterogeneity structured | | 1.87 (0.93to 2.99) | 1.15 (0.66 to 1.85) | 2.12 (1.30 to 3.38) | 0.84 (0.53 to 1.38) |
| $\phi$ (Decay of spatial correlation) | | 77.14 (38.77 to 99.23) | 84.45 (51.91 to 99.49) | 76.60 (25.71 to 99.17) | 76.81 (39.62 to 99.33) |
| Deviance Information Criterion | | 631.70 | 1996.00 | 1281.00 | 2506.00 |

CrI: 44.1–64.5%) lower risk of Wb123 Ab positivity than males. There was also an estimated increase of 2.4% (95% CrI: 1.8%–3.1%) in Wb123 Ab-positivity for every year of age (Table 3). Also, there was an increase in prevalence of being positive for Wb123 Ab of 125.6% (95% CrI: 37.4–276.6%) and 42.7% (95% CrI: 7.2–93.0%) for tuna cannery workers and those who work in other locations (excluding outdoors and tuna cannery), respectively, compared to indoor workers. Additionally, there was a significant increase of 0.2% (95% CrI: 0.1–0.5%) in the prevalence of Wb123 Ab-positivity for each 1% increase in the coverage of trees in the 20 m buffers.

The spatial Bm14 Ab model had a DIC of 1281, while the model without the spatial component had a DIC of 1898.6. In the spatial model, there was a decrease in the prevalence of Bm14 Ab of 52.1% (95% CrI: 35.7–64.9%) for females compared to males. Age was also as significant covariate with an increase in the prevalence of Bm14 Ab of 3.3% (95% CrI: 2.5–4.1%) per

every year of age. The prevalence of positive for Bm14 Ab was higher for those who worked in tuna cannery and outdoor locations compare to those working indoors. The increase in the prevalence was 155.8% (95% CrI: 40.3–362.2%) and 78.9% (95% CrI: 29.6–334.1%), respectively. Tree coverage had a significant positive association with positivity for Bm14 Ab, with an estimated increase of 0.01% (95% CrI: 0.001–0.4%) in Bm14 positivity for each 1% increase in tree coverage in the 20 m buffer area. Population density had a significant negative association with Bm14 Ab prevalence, with a decrease of 8.1% (95% CrI: 5.4–20.2%) for every person/m$^2$.

The spatial model for Bm33 Ab also had a lower DIC, 2506, compared with the nonspatial model, 3505.6. Similar to all the other infection markers, the decrease in prevalence of Bm33 Ab was 26.8% (95% CrI: 11.1–39.8%) in females compared to males, and the increase per every year of age was 2.4% (95% CrI: 1.8–3.03%). Also, workers in tuna cannery and outdoor locations had an increase of 56.2% (95% CrI: 3.3–150.1%) and 161.6% (95% CrI: 9.0–534.7%) compared to workers in indoor areas. The prevalence of Bm33 positivity was found to increase by 0.4% (95% CrI: 0.1–0.8%) with a 1% increase in the extent of tree coverage in the 20 m buffers.

In the model of Wb123 Ab the variance of the spatially structured random effect was 1.1 (0.7 to 1.8) and in the models of Bm14 and Bm33 these parameters were 2.1 (1.3 to 34) and 0.8 (0.5 to 1.4), respectively, meaning that the residual spatial variation was higher for the model of Bm14 Ab. The value of the decay parameter for spatial correlation ($\phi$) was 84.5 for Wb123 Ab, 76.6 for Bm14 Ab, and 76.8 for Bm33 Ab. These estimates indicate that after accounting for the effect of covariates, the radii of the clusters were approximately 3.9, 4.3 and 4.3 km, respectively.

## Spatial predictions

Maps of the mean and standard deviation (SD) of the posterior distributions of predicted probability of each of the LF infection markers are shown in Fig 4. The highest predicted probability of all infection makers (≥0.61) was mainly confined in the north-west part, an area that corresponds largely to the coastal villages of Fagali'i and Fagamalo. There were also predicted residual foci of high probability of Bm33 and Wb123 Abs in the southwest part of Tutuila, in areas that belong to Vaitogi and Futiga villages and in the western part of Tafuna village (0.21 and 0.49). High probability of Bm33 Ab covered larger areas compared to the other infection markers (≥0.21), with higher probability estimates in confined areas in the north-west (≥ 0.61), southwest (≥0.51), the north-east (≥0.51) and the central part around the Pago Pago area (≥0.41). The maps of the posterior SDs demonstrate that the level of uncertainty was higher in inhabited areas in the north that were predominantly covered by trees (Figs 2 and 4).

The models for Ag, Wb123 and Bm14 Abs were able to predict the probabilities of these outcomes reasonably well with AUC values of 0.71, 0.70 and 0.70, respectively. However, the model of Bm33 Ab performed poorly with an AUC value of 0.60 (S8 Fig). Bm33 Ab is the infection marker that is more widespread in American Samoa, and the poor performance of the model could be indicating that sociodemographic and environmental factors may not be important determinants of its distribution across the territory.

## Discussion

In this study, we conducted a Bayesian geostatistical analysis of LF infection markers at the household level and produced predictive probability maps for American Samoa in 2016. In addition, this study examined potential sociodemographic and environmental factors that may influence the geographical distribution of LF in the territory. To our knowledge, this is the first time that the distributions of LF infection markers have been examined at such high spatial resolution to predict LF probability. Our results suggest that there are still areas with high

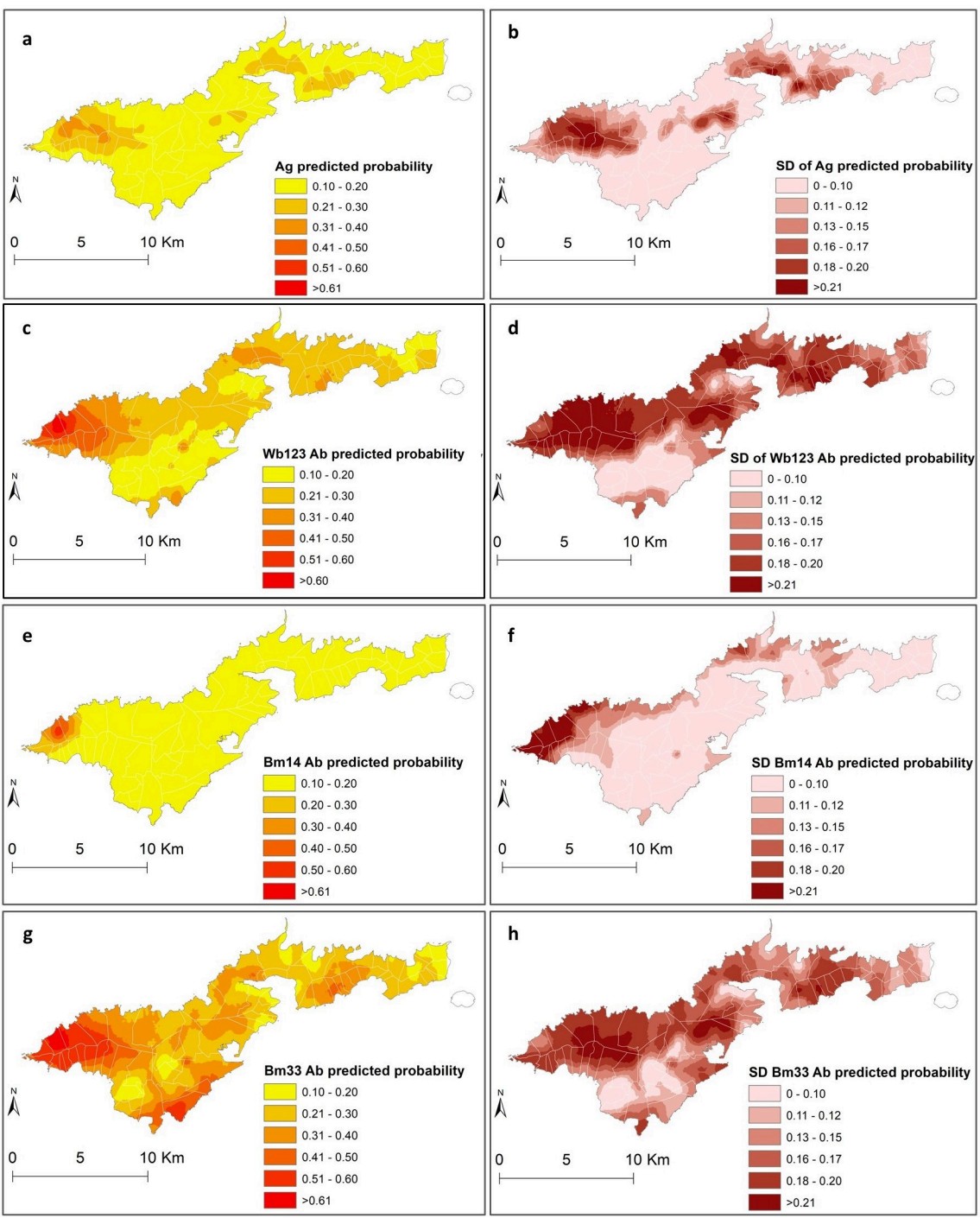

**Fig 4.** Spatial distribution of predicted probability and standard deviations of Ag (a and b), Wb123 Ab (c and d), Bm14 Ab (e and f), and Bm33 Ab (g and h) in American Samoa 2016. Base layer from: (https://www.diva-gis.org/gdata/).

probability of LF infection markers (including Ag) in American Samoa, particularly in the north-west of the main island of Tutuila. Also, we found that there are sociodemographic and environmental factors that may underly the geographical distribution of LF and potentially contribute to persistent transmission. These predicted probability estimates of LF infection

markers may help maximise the effectiveness of post-intervention surveillance by contributing to the identification of areas with highest probability of residual transmission [7].

The results showed that the predicted probability of Ag, Wb123, Bm14 and Bm33 Abs differed geographically across the territory. Areas around Fagali'i and Fagamalo villages in the north-west had the highest predicted probability of all infection markers. Also, in the south, high predicted probability, particularly for Bm33 Ab, were observed in localised areas in Vaitogi, Futiga and Tafuna villages. These findings concurred with the results of a previous research conducted in the territory that found significant spatial dependency for all infection markers, and confirmed the presence of LF clusters and hotspots in the north-west, south and central part of Tutuila [18]. The high-risk area in the far north-west was also previously identified as potential hotspot of residual infection in cross-sectional surveys conducted in American Samoa in 2010, 2014 and 2016 [8,10,11]. In this study, the cluster sizes for all infection markers were larger compared to the previous findings [8,18]. This discrepancy in cluster size may be explained by the implementation of different spatial methods, and also by the incorporation of sociodemographic and environmental covariates into the geostatistical models (noting that the cluster size is in the residual component). These covariates may be associated with heterogeneous exposure to mosquito bites. In areas where the parasite is transmitted predominantly by night-biting mosquitos, clustering of infection around household locations can be expected and has been demonstrated [30,32,47]. The results of this study support recent evidence that the home environment may be also an important area for exposure in LF-endemic regions where *W. bancrofti* is transmitted by the day-biting mosquito, *Ae. polynesiensis*. [48,49]. The cluster size also suggests that transmission may be occurring not only around households, but also in surrounding areas where the household members are likely to frequent (such as bus stops, schools and workplaces). In Samoa, a multilevel hierarchical modelling found that the intraclass correlation coefficients for Ag-positive individuals was higher at households (0.46) compared to primary sampling units (0.18) and regions (0.01) [49]. The timely identification of these small pockets of residual infection can be used to prioritise further interventions to reduce the risk of LF recrudescence or resurgence in the territory.

The predictive models developed for the different LF infection markers can help characterise the spatial patterns of serological responses to LF in American Samoa. In *W. bancrofti* endemic areas, WHO recommends the use of Ag testing to assess the impact of the MDA and determine when the elimination targets have been reached [16]. However, there is increasing evidence that suggests that the use of Ag alone for post-MDA surveillance may not be sufficiently sensitive to detect residual infection [7,50,51]. Therefore, antifilarial Ab testing is currently been examined as an alternative or complementary method of diagnosing LF in post-MDA surveillance surveys [17,50–52]. However, the dynamics of the Ab responses post-infection and post-treatment are still not well understood [53]. In this study, the geographical distribution of the predicted prevalence of Wb123 and Bm14 Abs were more clustered compared to the widespread distribution of positive Bm33 Ab responses. It has been observed that Abs can be detected earlier and are higher in prevalence when compared to Ag [54,55]. Also, Ab clearance is highly variable and can persist for years after treatment in some individuals [54]. This finding suggests that Bm33 Ab may not be the best indicator to identify areas of ongoing *W. bancrofti* transmission but may be used to provide information about levels of historical exposure and infection. This finding also concurred with a recent study indicating that the combined use of Ag and Bm14 Ab provides a more sensitive marker of current or previous infection than each indicator alone [56]. Additional longitudinal studies are required to help monitor how the stage of the infection and magnitude of the immunological responses determine the spatial patterns of antifilarial Abs. Such information will have implications for the selection of the most suitable LF diagnostic tools in low prevalence and post-MDA settings.

There were consistent associations between the infection markers and the sociodemographic variables included in the models. The observed differences among females and males and the positive association with age is most likely to be exposure-related. These findings support what has been observed previously in the territory and in most LF-endemic areas [8,48,57]. Males spend more time working outdoors compared to females. However, it has also been suggested that immunological and hormonal gender differences may account for the lower infection rates in females [58]. There was a consistent positive association between all Abs and tuna cannery workers. Also, a positive association between Bm14 and Bm33 Abs and individuals working in outdoor locations. In 2013, An average of 17.6% of the total employed population in the territory worked in tuna cannery which is the largest non-government employer in American Samoa [59]. Tuna cannery workers are typically of low socioeconomic status and their poor living conditions may increase their risk of exposure to the infection. Higher prevalence of Wb123 Ab was also previously observed in tuna cannery workers in the territory but no associations between Ab responses has been identified with other occupational groups [8].

The spatial models for all infection makers indicated that there was a positive association between the prevalence of LF and the extent of tree coverage in the 20 m buffers. This finding supports the hypothesis that the tree coverage may impact mosquito population dynamics and behaviours [60]. Tree canopy may sustain *W. bancrofti* life cycle in high temperature areas by facilitating the survival of mosquitos that move in response to food supply [12]. Most of American Samoa is steep, with approximately half of the area covered by rainforest [12,61]. Trees primarily cover most areas in the northern part of the territory (Fig 2) where the highest prevalence of LF was observed. Rainfall has been shown to be associated with high prevalence of LF in several endemic countries where the infection is transmitted by different vectors [20,23,31,62,63]. Rainfall generates water pools that can serve as mosquito breeding sites influencing mosquito abundance and behaviour [64]. In contrast, heavy rainfall may have a negative effect on LF prevalence by causing excessive damage to mosquito larval habitats [64]. No associations between the prevalence of infection markers and rainfall were found in this study. This finding was unexpected and deserves further investigation. These findings raise the need for high-quality spatial environmental datasets that can be used in further studies to determine the association of LF and other potential environmental drivers.

The strengths of this study include the availability of data at the household level that allowed us to assess the geographical distribution of LF in American Samoa at a small spatial scale. In this way, it was possible to explore the home environment as an exposure area of importance. The study also developed geostatistical models for different infection markers that may be used as baseline information to characterise the spatial patterns of the antifilarial Ab responses in the long-term. Besides the predicted prevalence maps, the spatial models developed here also provided outputs to determine the associated uncertainty of the prevalence estimates [65]. The maps of the SD (uncertainty) highlight the areas where predictions were imprecise and that need to be explored in future studies.

The limitations of the study include the lack of high-quality spatial environmental datasets for the territory. As a result, it was not possible to include covariates such as temperature, that has consistently been associated positively with LF [19,20,31]. Also, the rainfall data used in the study was only available in a spatial format for the year 2016. Based on the assessment presented in the supplementary files (S1 Table and S1 and S2 Figs), data were found to be representative of the average rainfall estimates for the ten-year period prior the survey (most likely time period of potential exposure). Despite this limitation, we believe that our results provide valuable information about the potential sociodemographic and environmental factors that may be influencing the distribution of the infection in American Samoa. The geographical

distribution of LF may be influenced by the effect of the environment on vector mosquito populations and dynamics. A previous study conducted in American Samoa at the village level found statistically significant associations between PCR-positive *Ae. polynesiensis*, the primary LF parasite vector in the territory, and human seroprevalence of Ag and Wb123 Ab [9]. Mosquito data at the household level were no available for this analysis. Therefore, further interventions specifically designed to collect higher resolution data on both, humans and mosquitos, are needed to assess their spatial distributions and associations with environmental and sociodemographic covariates.

In this study, the Bayesian geostatistical models incorporating sociodemographic and environmental covariates showed that the predicted prevalence of LF was not homogeneous in American Samoa. Small-scale spatial variation in LF prevalence was observed which indicates that there is scope for further spatial analyses to help inform spatially-targeted interventions in American Samoa. Areas of priority for further study include the north and south-western part of the territory. Also, longitudinal monitoring of the prevalence of Ag, Wb123, Bm14 and Bm33 Abs would be useful to better understand the dynamics and potential use of different LF infection markers to inform and support the ongoing post-MDA surveillance efforts.

## Supporting information

**S1 Table. Average monthly rainfall (mm) in the Pago Pago area, in American Samoa from 2000-2020.** Data extracted from the National Weather Services Websiteand average month rainfall from the spatial layer in 2006.
(DOCX)

**S2 Table. Odds ratios (ORs) and 95% CrI from the non-spatial models for Ag and Wb123, Bm14 and Bm33 antibodies in a community survey in American Samoa in 2016.**
(DOCX)

**S1 Fig. Average monthly rainfall (mm) for the period 2000–2020 and average montly rainfall (mm) in 2016 in American Samoa.** The driest (August) and wettest (December) months in 2016 were representative of the average rainfall in the respective months in previous 20 years.
(DOCX)

**S2 Fig. Average annual rainfall (mm) for the period 2000–2020.** The horizontal line indicates the average in the previous 10 years. Total rainfall in 2016 was representative of average rainfall in the previous 20 years.
(DOCX)

**S3 Fig. Correlation matrix of the variables extracted for the analysis.**
(DOCX)

**S4 Fig. History plots of the last 6,000 values from the posterior distributions of the covariate parameters in the model for Ag.**
(DOCX)

**S5 Fig. History plots of the last 10,000 values from the posterior distributions of the covariate parameters in the model for Wb123 Ab.**
(DOCX)

**S6 Fig. History plots of the last 10,000 values from the posterior distributions of the covariate parameters in the model for Bm14 Ab.**
(DOCX)

**S7 Fig. History plots of the last 10,000 values from the posterior distributions of the covariate parameters in the model for Bm33 Ab.**
(DOCX)

**S8 Fig. The area under curve (AUC) statistic of the receiver operating characteristic curve for (a) Ag, (b) Wb123, (c) Bm14 and (d) Bm33 Abs.**
(DOCX)

**S1 Text. OpenBuGS code of Bayesian geostatistical models for the probability of being positive for antigen in American Samoa, 2016.** All of the parameters in this code were incorporated in the models for Wb123, Bm14 and Bm33 Ab.
(DOCX)

## Author Contributions

**Conceptualization:** Angela M. Cadavid Restrepo, Colleen L. Lau.

**Formal analysis:** Angela M. Cadavid Restrepo, Colleen L. Lau.

**Writing – original draft:** Angela M. Cadavid Restrepo, Colleen L. Lau.

**Writing – review & editing:** Angela M. Cadavid Restrepo, Beatris M. Martin, Saipale Fuimaono, Archie C. A. Clements, Patricia M. Graves, Colleen L. Lau.

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
