## [Decision Letter · Decision Letter 0]

28 Nov 2022

Dear Dr. Cadavid Restrepo,

Thank you very much for submitting your manuscript "Predictive risk mapping of lymphatic filariasis residual hotspots in American Samoa using demographic and environmental factors" for consideration at PLOS Neglected Tropical Diseases. As with all papers reviewed by the journal, your manuscript was reviewed by members of the editorial board and by several independent reviewers. In light of the reviews (below this email), we would like to invite the resubmission of a significantly-revised version that takes into account the reviewers' comments. 

We cannot make any decision about publication until we have seen the revised manuscript and your response to the reviewers' comments. Your revised manuscript is also likely to be sent to reviewers for further evaluation.

Sincerely,

Song Liang

Academic Editor

Francesca Tamarozzi

Section Editor

Editors feedback

Please note that PLoS NTD requires authors to make all data underlying their work publicly available - either within the submission files, or via a data repository. Please, within the resubmission, include raw data or link to an open-access repository, for data to be available to the Editors and the Reviewers. Handling of requests to access original data files directly by the authors is not allowed as per PLoS NTD policy. 

In case of concerns to share the data publicly (e.g. to protect patient confidentiality), please do get in touch as soon as possible for further guidance. 

Reviewer's Responses to Questions

**Key Review Criteria Required for Acceptance?**

**Methods**

-Are the objectives of the study clearly articulated with a clear testable hypothesis stated?

-Is the study design appropriate to address the stated objectives?

-Is the population clearly described and appropriate for the hypothesis being tested?

-Is the sample size sufficient to ensure adequate power to address the hypothesis being tested?

-Were correct statistical analysis used to support conclusions?

-Are there concerns about ethical or regulatory requirements being met?

Reviewer #1: L310: Separate models were fitted to different infection markers. Is it possible to just fit one model using the true prevalence rate as a latent variable and the prevalence rate of these four infection markers as different sets of observations of it. This is just a suggestion, no need to incorporate in the current manuscript.

L313-6: These sentences are confusing. Am I understanding correctly that L312-4 describes the non-spatial model and L314-6 describe the spatial model, and the only difference is if the spatial random effect sj is included? I think it’s not fair to refer the model as “geostatistical model” if the spatial term is not included. The sentences could be modified to something like “We first developed a non-spatial model with the sociodemographic….. as the fixed effects, then extended this model to be a spatial model by including a spatial random effect term sj (Equation 1).”

L316-18: The statement about MCMC does not feel correct. Other methods, such as Restricted maximum likelihood can also accomplish these.

L331: (1) Why assuming a linear relationship between age and the prevalence rate? (2) Since you already have household information, why not including it in the model? The infection status of people living in the same household are unlikely to be independent. (3) Several variables are included for occupation explicitly. What the reference level is? If Occupation is included in the model as a categorical variable, this should be done implicitly by the regression function when generating the design matrix. Moreover, the categories mentioned here are different from those presented in Table 3.

L351: The convergence plots could be added to the appendix.

L364: What’s the purpose for using Bayesian kriging? Just for visualizing the results at a finer scale? If so, I think it is unnecessary given the low printing resolution anyway.

Reviewer #2: - The study objectives are clearly stated.

- The population and sample size need to be clarified in the method section.

- Please clarify the description of Fig 2, what is K x 0.02 (line 276)

-It is not clear to me why the author excluded the temperature data from analyses. The author used rasters of rainfall and temperature with the same resolution at 1 km but it turns out that temperature data wet not available for large areas (lines 392-393).

Reviewer #3: “Predictive” seems to be a misnomer in the title. The bulk of the modeling is association-based, with the spatial prediction occurring with only environmental data/population density at unsampled points. Crucially, no model validation was conducted. Why? The author could use a subset of sampled households/individuals and test on the remainder. Instead, prediction is made on unsampled continuous space. How much of that is inhabited? The authors never provide justification for this approach, as the Bayesian kriging is mostly reproducing the patterns from survey results.

Line-specific comments:

326: Talk about Bernoulli assumptions

Fig 4: Methods of binning for SD plots (b,d,f,h) are not uniform. Please make comparable and justify

**Results**

-Does the analysis presented match the analysis plan?

-Are the results clearly and completely presented?

-Are the figures (Tables, Images) of sufficient quality for clarity?

Reviewer #1: L374-7: Why the prevalence rates vary remarkably across infection markers?

L381-4: Are the prevalence rates by infection markers highly correlated? What are the distributions of them? The fact that “villages with high prevalence of Bm33 Ab were more widespread” might due to that the prevalence rate of Bm33 Ab are higher than other places in general, and it looks like it’s high every where in Fig 3f when using the color scale that is appropriate for the other infection markers. 

L394: A corrplot could be included to represent the correlation coefficients between variables. https://cran.r-project.org/web/packages/corrplot/vignettes/corrplot-intro.html

Table 2: I think this table could be moved to the appendix since it does not provide any necessary information for understanding the model or interpretating the model results.

Table 3: What does the blue background refers too? Looks like it represents cells with a significant relationship. Then why not use different colors for positive and negative relationships.

Table 3&4: If spatial model always performs better, I think it is reasonable to just include the results for spatial model in the main text and put the results for the non-spatial model in the appendix. 

L419-22: I feel interpreting the spatial scale parameter as the size of cluster is unreasonable. It stands for the cluster size of uncontrolled confounders

Reviewer #2: - Fig 1. Please check the label of the elevation classification in the legend. The author used classes 0 and 0-100.

- Fig 3 a) needs to be expanded to show the locations of sample villages. The author mentions many times the high prevalence in the northwest, particularly Fagali'i and Fagamalo. It would be more clear for the readers if you move Fig 3a) to the study area section and make it bigger. 

- Please check the numbers used in the paragraph (Pages 21-23) and Table 3.

Reviewer #3: Additionally and crucially, on the prediction part, the values found and displayed in Figure 3 (survey data) and Figure 4 (prediction) are two orders of magnitude apart. Why? This seems to be more than just a rescaling error, as the text includes % that confirm this difference. It is hard to square this difference.

Figures 2-4 as provided are too low resolution.

**Conclusions**

-Are the conclusions supported by the data presented?

-Are the limitations of analysis clearly described?

-Do the authors discuss how these data can be helpful to advance our understanding of the topic under study?

-Is public health relevance addressed?

Reviewer #1: (No Response)

Reviewer #2: - Page 32, please clarify the association between the prevalence and tree coverage. It seems like the results show that 95% CI of all ORs include 1.00 (Table 3 and 4)

Reviewer #3: There are some major gaps in interpretation that are never addressed. The meaning of Ag vs. Wb123Ab vs. Bm14Ab vs. Bm33Ab are never satisfactorily addressed and should be brought up in the Introduction since Results interpretation depends on knowing their differences. I understand there is literature elsewhere on this, but a paper should stand-alone to a certain degree, and this is never addressed herein.

The role of the mosquito is also not well described or addressed in this paper, as they are strongly influenced by the environmental layers and other covariates used in the models. Why not? If there is no data on mosquitoes, then say so but explain what it would contribute; if there is, it should be incorporated into the modeling.

On both of these points, contradictory discussion is offered. The authors comment that there is “widespread distribution of positive Bm33 Ab responses”, but then can go on to say that :Bm33Ab can be detected more than a year before other Ab responses and can decrease after MDA”. Which one is more influential on results interpretation? They seem to point in opposite directions. Re:mosquitoes, there is brief suggestion of the importance of day-biting and night-biting mosquitoes but it’s not clear which is more important in American Samoa (or if they matter in different areas). There is also a line about risk to tuna cannery workers based on time of day when they work, but that is never explained (is it during the day? at night more?).

Other comments:

Why was rainfall non-association finding unexpected? Never explained.

**Editorial and Data Presentation Modifications?**

Reviewer #1: (No Response)

Reviewer #2: Minor Revision

Reviewer #3: 123: Ag not defined yet but abbreviation used

126: What is the threshold?

130: What to make of other studies showing ongoing transmission while TAS-1 and TAS-2 were passed? Please comment.

153-155: Sentence fragment, rewrite

Fig 2: Fix population density units label, convert temperature map to something more human readable (Celsius perhaps)

301-3: Rephrase this sentence (verb agreement)

331: Starkist? Should you just say (Tuna) Cannery?

page 30, etc.: Lowercase for species names (polynesiensis, bancrofti)

**Summary and General Comments**

Reviewer #1: This manuscript developed a Bayesian geostatistical model to examine the environmental determinants of lymphatic filariasis and predict its prevalence on a fine scale for American Samoa. In general, I think the method is valid and the results are well-presented. Some clarifications could improve the readability further. For example, what do the four infection markers represent and what are the relationships between them? Do they refer to infections caused by different types of parasites and are there systematic relationships between them?

Reviewer #2: The Bayesian model-based geographics approach combines with sociodemographic and environmental

 data were used to predict the spatial distribution of LF prevalence in American Samoa. The manuscript was well written and will make a good contribution to the knowledge of LF epidemiology in American Samoa.

Reviewer #3: The motivating question for this paper is interesting; how do we use modeling to support more specific targeting of post-MDA surveillance activities? I don’t think the methods are sufficient though and the claim of being “predictive” is a misnomer. Model validation is a must; the orders-of-magnitude gulf between surveys and predicted values is not reassuring to readers. I would encourage the authors to rethink their approach.

PLOS authors have the option to publish the peer review history of their article (what does this mean?). If published, this will include your full peer review and any attached files.

Reviewer #1: Yes: Qu Cheng

Reviewer #2: No

Reviewer #3: No
---

## [Decision Letter · Decision Letter 1]

29 Mar 2023

Dear Dr Cadavid Restrepo,

Thank you very much for submitting your manuscript "Spatial predictive risk mapping of lymphatic filariasis residual hotspots in American Samoa using demographic and environmental factors" for consideration at PLOS Neglected Tropical Diseases. As with all papers reviewed by the journal, your manuscript was reviewed by members of the editorial board and by several independent reviewers. In light of the reviews (below this email), we would like to invite the resubmission of a significantly-revised version that takes into account the reviewers' comments. 

The reviewers have appreciated the improvement of the manuscript, but one reviewer in particular expressed still concern on the validation of the model.

Furthermore, PLoS NTD policy requires fulla availability of data. We do understand that the study place is small and even anonymized data can contain enough information for people to be possibly identified, but further conceiling of such data can be made, for example change the village name with letter (village A, B and so on). Also, full methods data must be available.

We cannot make any decision about publication until we have seen the revised manuscript and your response to the reviewers' comments. Your revised manuscript is also likely to be sent to reviewers for further evaluation.

Sincerely,

Francesca Tamarozzi

Section Editor

Francesca Tamarozzi

Section Editor

The reviewers have appreciated the improvement of the manuscript, but one reviewer in particular expressed still concern on the validation of the model.

Furthermore, PLoS NTD policy requires fulla availability of data. We do understand that the study place is small and even anonymized data can contain enough information for people to be possibly identified, but further conceiling of such data can be made, for example change the village name with letter (village A, B and so on). Also, full methods data must be available.

Reviewer's Responses to Questions

**Key Review Criteria Required for Acceptance?**

**Methods**

-Are the objectives of the study clearly articulated with a clear testable hypothesis stated?

-Is the study design appropriate to address the stated objectives?

-Is the population clearly described and appropriate for the hypothesis being tested?

-Is the sample size sufficient to ensure adequate power to address the hypothesis being tested?

-Were correct statistical analysis used to support conclusions?

-Are there concerns about ethical or regulatory requirements being met?

Reviewer #1: (No Response)

Reviewer #2: Do the study area information original from the author, if not please add appropriate references. 

Line 220: The author stated that a sample size was 4620, how the author calculated the sample size? Why the sample size and number of participants are difference? 

Please describe how many household members aged>8 years in 32 PSUs and 754 households in total and how many of them were surveyed?

Reviewer #3: The manuscript is improved from the earlier draft in that it attempts to perform model validation. However, what is described is basic and not sufficient. Cross validation is needed rather than removing 25% of the data one time.

Why was using intervention terms in the model not investigated or discussed?

I am not sure why the sample size information is included when it was not attained to reach the power pre-specified. Is this a problem for this paper rather than the prior studies cited? Nonetheless, I do not dispute its inclusion since another reviewer requested it, but I do not see its added value.

**Results**

-Does the analysis presented match the analysis plan?

-Are the results clearly and completely presented?

-Are the figures (Tables, Images) of sufficient quality for clarity?

Reviewer #1: (No Response)

Reviewer #2: I have no further comment in the result section

Reviewer #3: The AUC values are not compelling, and I would like to see it done with cross validation, including reporting of variance. Three markers are around 0.7 (the threshold chosen by the authors but not an impressive one), and another is close to 0.6. If the best models are still not performing well, that is okay to report and publish, but the text/results on AUC are key to the paper and haven't been appropriately treated in this revision.

S4-S7: For these plots, it is not clear by visual inspection that convergence is achieved. Are these the wrong 10,000 values shown? You are fitting a lot of parameters. How do we know to not be concerned about overfitting? Please check.

**Conclusions**

-Are the conclusions supported by the data presented?

-Are the limitations of analysis clearly described?

-Do the authors discuss how these data can be helpful to advance our understanding of the topic under study?

-Is public health relevance addressed?

Reviewer #1: (No Response)

Reviewer #2: (No Response)

Reviewer #3: The model results (significant terms, DIC, AUC, etc.) vary quite a bit by antibody, but the authors do not spend significant time/text discussing this or thinking about the interpretation of results. A reviewer from the prior draft also brought this up, and the authors have not adequately addressed this point despite starting to sketch out some important points in their response without modifying the text. To get to public health relevance, the relationships need to be understood, and model results should at least accord with biological understanding, or commentary should be made about the mismatch.

**Editorial and Data Presentation Modifications?**

Reviewer #1: (No Response)

Reviewer #2: (No Response)

Reviewer #3: Conversion of cluster size inexact (line 483) because American Samoa is not at equator. Please use a better method.

S4: Why only 6,000 values instead of 10,000? Please correct.

**Summary and General Comments**

Reviewer #1: (No Response)

Reviewer #2: (No Response)

Reviewer #3: The manuscript is improved from the prior draft but still needs major improvements related to cross validation, checking convergence and overfitting, and interpretation of model results for public health relevance meaningfully for readers.

Data should be made publicly available so results can be replicated. Ideally, code would also be shared so reviewers/future readers can have confidence in the results and conclusions presented herein.

PLOS authors have the option to publish the peer review history of their article (what does this mean?). If published, this will include your full peer review and any attached files.

Reviewer #1: Yes: Qu Cheng

Reviewer #2: No

Reviewer #3: No
---

## [Editor Report · Decision Letter 2]

26 Jun 2023

Dear Dr. Restrepo,

We are pleased to inform you that your manuscript 'Spatial predictive risk mapping of lymphatic filariasis residual hotspots in American Samoa using demographic and environmental factors' has been provisionally accepted for publication in PLOS Neglected Tropical Diseases.

Best regards,

Song Liang

Academic Editor

Francesca Tamarozzi

Section Editor

---

## [Editor Report · Acceptance letter]

20 Jul 2023

Dear Dr Cadavid Restrepo,

We are delighted to inform you that your manuscript, "Spatial predictive risk mapping of lymphatic filariasis residual hotspots in American Samoa using demographic and environmental factors," has been formally accepted for publication in PLOS Neglected Tropical Diseases.

Best regards,

Shaden Kamhawi

co-Editor-in-Chief

Paul Brindley

co-Editor-in-Chief
